# Study of the Sensing Kinetics of G Protein-Coupled Estrogen Receptor Sensors for Common Estrogens and Estrogen Analogs

**DOI:** 10.3390/molecules28083286

**Published:** 2023-04-07

**Authors:** Dingqiang Lu, Xinqian Wang, Chunlei Feng, Danyang Liu, Yixuan Liu, Yujiao Liu, Jie Li, Jiayao Zhang, Na Li, Yujing Deng, Ke Wang, Ruijuan Ren, Guangchang Pang

**Affiliations:** 1College of Biotechnology & Food Science, Tianjin University of Commerce, Tianjin 300134, China; 2Tianjin Key Laboratory of Food Biotechnology, Tianjin 300134, China; 3Tianjin Institute for Food Safety Inspection Technology, Tianjin 300134, China

**Keywords:** GPER, estrogens, receptor-ligand interaction, interconnect allosteric constant, molecular simulation docking, electrochemical signal amplification system

## Abstract

Endogenous and exogenous estrogens are widely present in food and food packaging, and high levels of natural estrogens and the misuse or illegal use of synthetic estrogens can lead to endocrine disorders and even cancer in humans. Therefore, it is consequently important to accurately evaluate the presence of food-functional ingredients or toxins with estrogen-like effects. In this study, an electrochemical sensor based on G protein-coupled estrogen receptors (GPERs) was fabricated by self-assembly, modified by double-layered gold nanoparticles, and used to measure the sensing kinetics for five GPER ligands. The interconnected allosteric constants (*K*_a_) of the sensor for 17β-estradiol, resveratrol, G-1, G-15, and bisphenol A were 8.90 × 10^−17^, 8.35 × 10^−16^, 8.00 × 10^−15^, 5.01 × 10^−15^, and 6.65 × 10^−16^ mol/L, respectively. The sensitivity of the sensor for the five ligands followed the order of 17β-estradiol > bisphenol A > resveratrol > G-15 > G-1. The receptor sensor also demonstrated higher sensor sensitivity for natural estrogens than exogenous estrogens. The results of molecular simulation docking showed that the residues Arg, Glu, His, and Asn of GPER mainly formed hydrogen bonds with -OH, C-O-C, or -NH-. In this study, simulating the intracellular receptor signaling cascade with an electrochemical signal amplification system enabled us to directly measure GPER–ligand interactions and explore the kinetics after the self-assembly of GPERs on a biosensor. This study also provides a novel platform for the accurate functional evaluation of food-functional components and toxins.

## 1. Introduction

Estrogens are fat-soluble steroid hormones with important roles in human development, cellular differentiation, maturation, reproduction, and diseases [1]. Environmental estrogens (EEs) affect the normal functioning of organisms by interfering with the actions of endogenous estrogens. EEs include exogenous estrogens such as nonylphenol (NP), bisphenol A (BPA), resveratrol (Res), and daidzein [2], in addition to endogenous estrogens in environments such as 17β-estradiol (E2) and estriol [3]. EEs are an important class of environmental endocrine disruptors and almost all EEs can interact with estrogen receptors (ERs), resulting in biological effects [4]. ERs fall into two broad categories including nuclear estrogen receptors (e.g., ERα and ERβ) and G protein-coupled estrogen receptors (GPERs) that respectively mediate the physiological functions of estrogens through genetic and non-genetic pathways [5]. Nuclear estrogen receptors take hours or even days to regulate gene expression; therefore, estrogen-like food-functional ingredients or toxins likely act through fast signal transduction pathways mediated by GPERs [6]. After estrogens interact with the ligand-binding domains of GPER, the receptor couples to intracellular G proteins through complex conformational changes of seven transmembrane domains and cascade reactions and the processes selectively activate multiple downstream signaling cascade amplifications, resulting in various biological effects [6,7]. Studies have demonstrated that many EEs are associated with GPERs, but GPER-mediated estrogen interference effects have not received much research attention [2].

E2 was used in this study among the various ligands of GPER, because it is the most abundant and active natural estrogen [2], in addition to being the most studied natural GPER ligand that features a binding affinity of 3–6 nmol/L [8]. Phytoestrogens with two phenol rings are nonsteroids with structures like E2 and can interact with ERs in vivo resulting in weak estrogen-like effects. Consequently, phytoestrogen resveratrol was selected since it is a “star” monomer of stilbenes that binds to ERs and exhibits a strong estrogen-like activity [9]. The exogenous estrogen BPA was also selected because of its estrogen-like effect and because it is frequently used in the manufacturing process of food packages, medical devices, milk bottles, spectacle lenses, and many other daily necessities. Further, BPA is easily absorbed by humans and can cause endocrine disorders, malformations, and cancers [10]. The most widely used GPER-specific agonist, G-1 [11], and the GPER-specific antagonist, G-15 [12], were also selected as ligands for determination and comparisons.

Recent studies of receptor–ligand interactions have focused on the strength of the affinity of ligands to the extracellular domains of receptors using sensors based on surface plasmon resonance (SPR), photoelectrochemistry (PEC), and surface-enhanced Raman spectroscopy (SERS). These methods directly measure receptor–ligand interactions in-dependent of binding sites, probes, or signaling pathways, and they cannot measure the intracellular signaling cascades triggered by ligand interactions with the extracellular domains of receptors [13]. Cytology-based methods are also widely used for studying receptor–ligand interactions. Such studies investigate physiological activities of an organism through physiological changes of cells caused by receptor–ligand interactions including changes in nerve impulse signals resulting from ion channel modification. However, these physiological changes are easily affected by cell types, cell–cell interactions, complex intracellular signaling pathways, and their “cross-talk”, therefore resulting in a poor reflection of true physiological activities of organisms [14]. Our previous studies of biosensors [15,16], including tissue sensors and especially receptor sensors [17,18,19,20], indicated promising results via the fabrication of G protein-coupled receptor (GPCR)-based nano and tissue sensors, receptor gene expression, self-assembly, cross-species self-assembly sensors, and the quantification and kinetics analysis. The accurate evaluation of food-functional ingredients or toxins that exhibit estrogen-like effects is an area of current research focus. GPERs can be used as a standardized measurement platform for quantitatively analyzing the functions of food-functional ingredients or toxins that have estrogen-like effects, thereby revealing new insights without interference from the cross-talk between signaling pathways in living-cell systems. In this study, an electrochemical signal amplification system was designed using gold nanoparticles (AuNPs) and horseradish peroxidase (HRP) to mimic the signaling cascades triggered by GPERs in cells. Further, a double-layer AuNP-modified GPER sensor was developed [21]. The kinetic parameters in the detection of five separate GPER ligands (E2, Res, BPA, G-1, and G-15) were quantitatively determined using the GPER double-layer gold nanoparticle receptor sensor on an electrochemical workstation. Revealing the kinetics and pertinent parameters of different food-functional components or toxins can play an important role in food-functional evaluation.

## 2. Results and Discussion

### 2.1. Expression of hGPER Receptor

Recombinant plasmids of PcDNA3.1(+)-GPER were transfected into HEK-293T cells. After being incubated for 48 h, all-protein extraction was performed and molecular weight was estimated by protein blotting. After transfection, the target gene was expressed in the cells as shown in Figure 1, and its molecular weight was about 42 kDa, which was consistent with the theoretical molecular weight [22], although the cells not transfected with the target gene plasmid did not express the protein, which indicated that the target gene plasmid could express GPER in HEK-293T cells.

### 2.2. UV-Vis and Transmission Electron Microscope (TEM) Characterization of AuNPs

The transparent burgundy-red GNPs prepared in this study were analyzed by spectral scanning in the wavelength range of 400–700 nm and TEM spectroscopy. From Figure 2A, the maximum spectral absorption value of the prepared GNPs was 519 nm, the absorbance of the prepared GNPs was 1.001, and the corresponding average particle size was 15 nm. From Figure 2B, it can be seen that the nanoparticles (GNPs) had an average particle size of about 15 nm, a regular spherical shape, uniform particle size, and no obvious aggregation phenomenon. The UV-Vis characterization and transmission electron microscopy scans of GNPs are consistent, which indicates that the produced nanoparticles have a size of 15 nm and can be well used for subsequent studies.

In addition, Figure 2A shows well the adsorption effect of GNPs on HRP. The black curve is the absorption spectrum scan of GNPs with a large absorption peak at 519 nm (peak 1); the blue curve is the absorption spectrum scan of HRP with an absorption peak at 405 nm (peak 2); and the red curve is the absorption spectrum scan of HRP adsorption by GNPs with an absorption peak at 405 nm and absorption peaks at 522 nm (peaks 3 and 4). Absorption peak 4 is red-shifted by 3 nm compared with peak 1, and the absorption value is also significantly lower, indicating that the GNPs adsorbed HRP. Absorption peak 3 is decreased compared with peak 2, but the maximum absorption wavelength is not changed: both are 405 nm, indicating that the adsorption of GNPs-fixed HRP has no effect on the structure of HRP and can keep the biological activity of HRP well.

### 2.3. Electrode Assembly and Characterization

#### 2.3.1. Electrode Pretreatment and Characterization

Cyclic voltammetry has become a fundamental sensing experiment method in the electrochemical analysis of receptor and tissue sensors [15,16,17,18,19,20]. Activation of a glassy carbon electrode (GCE) by H_2_SO_4_ generates negatively charged oxygen-containing groups (e.g., hydroxyl and carboxyl groups) on the electrode surface [23], leading to the formation of a porous structure that increases the effective surface area [24].

The GCE was characterized by cyclic voltammetry before and after pretreatment (Figure 3A). The redox peak current significantly increased after pretreatment, while the peak potential difference was less than 80 mV, and the peak current ratio was close to 1, indicating that the electrode met the experimental requirements. The cyclic voltammograms at different scan rates (25, 50, 75, 100, 125, 150, and 200 mV/s, scan range of −0.2 to 0.4 V) are shown in Figure 3B. A linear correlation was observed between the redox peak current and the square root of the scan rate, indicating that the peak current was only controlled by diffusion (Figure 3C). Thus, the GCE pretreatment was successful and H_2_SO_4_ activation increased electrode performance. The pretreated GCE could consequently be used for subsequent experiments.

#### 2.3.2. Electrochemical Characterization of the Assembly Process of Double-Layer GPER Nanogold Receptor Biosensors

Characterization of each stage of electrode assembly was performed with cyclic voltammetry using a scan rate of 50 mV/s and a scan range of −0.2 to 0.4 V in a K_3_Fe(CN)_6_ solution (1 × 10^−3^ mol/L, containing 0.20 mol/L KNO_3_) (Figure 4). After assembling chitosan (Chit) on top of GCE (curve b), the redox peak current was greatly reduced compared with GCE alone (curve a) due to the blockage of electron transfer by the Chit film, indicating successful assembly of Chit. The assembly of AuNPs on top of Chit (curve c) was successful based on the sharply increased redox peak current owing to accelerated electron migration by AuNPs. After assembling the GPER protein on top of AuNPs (curve d), the redox peak current was slightly reduced compared with that in curve c. This change could be due to steric hindrance of electron transfer by the assembled GPER, indicating the successful assembly of the receptor protein. After the assembly of thionine–chitosan (Thi-Chi) on top of the GPER protein (curve e), the redox peak current was elevated compared with curve d. The promotion of electron transfer by Thi outweighing blocked electron migration by Chit, indicating that Thi-chitosan was successfully assembled. After assembling AuNP-HRP on top of Thi-Chi (curve f), the peak current dramatically increased because of the redox catalytic activity of HRP (horseradish peroxidase) and the excellent electron transfer ability of AuNPs. Assembly of the second GPER protein layer on top of AuNP-HRP (curve g) blocked electron transfer and significantly reduced the peak current compared with curve f. Finally, bovine serum albumin (BSA) was used to block non-specific sites on the electrode surface (curve h), leading to a reduced peak current compared with curve g. Thus, the electrochemical double-layer AuNP-modified GPER sensor was successfully assembled.

### 2.4. Potential Optimization by the Current–Time Determination Method

The current–time method was used to measure the GPER electrochemical receptor sensor under different potential conditions. Ultrapure water was used as a blank control, and 1 × 10^−6^ mol/L E2 was added to the experimental group. The steady-state current change rate was measured before and after to measure the effect of different potentials on the response effect of this electrochemical receptor sensor. The results showed that the rate of change of the current was greatest at −0.38 V. Therefore, a constant potential of −0.38 V was selected for the study of the response characteristics of the biosensor to GPER ligand compounds.

### 2.5. Kinetic Determination of Estrogens and Estrogen Mimics with the Electrochemical Double-Layer AuNP-Modified GPER Sensor

#### 2.5.1. Determination of Response Ranges of Estrogens and Estrogen Mimics to GPER

The assembled electrode was used for measurement based on a time–current curve with the logarithm of ligand concentration, lg(C), taken as the x-axis, and the change rate of response currents (ΔI) was taken as the y-axis. E2 is shown in Figure 5(A1) as an example, wherein ΔI revealed an optimal linear correlation with increasing E2 concentration within the concentration range of 10^−16^ to 10^−14^ mol/L. Measurements of other ligands are shown in Figure 5(B1–E1).

#### 2.5.2. The Interconnected Allosteric Constant (Ka) Produced by Estrogens and Estrogen Mimics to GPER

Different concentrations of GPER ligands could be further subdivided into corresponding concentration ranges (Figure 5(A1–E1)). For example, the linear range (10^−16^ to 10^−14^ mol/L) in Figure 5(A1) was further subdivided and shown in Figure 5(A2), with ligand (estrogens and estrogen mimics) concentrations plotted on the x-axis and ΔI on the y-axis.

According to the receptor–ligand binding kinetics equation:(1)R+LRL←K2→K1
when the receptor is saturated,
(2)$$K_{d}=\frac{k_{2}}{k_{1}}=\frac{\left[R\right]\left[L\right]}{\left[RL\right]}$$

Let [*RT*] be the initial concentration of the receptor, then [*R*] = [*RT*] − [*RL*]. Let [*LT*] be the total ligand-based concentration, then [*L*] = [*LT*] − [*RL*]. Substituting [*L*] = [*LT*] − [*RL*] and [*R*] = [*RT*] − [*RL*] into Equation (2), we obtain, after finishing,
(3)$${\left[RL\right]}^{2}-\left[RL\right]\left\{\left[RT\right]+\left[LT\right]+K_{d}\right\}+\left[RT\right]\left[LT\right]=0$$

The above equation shows a hyperbolic quadratic equation with [*RL*] as the variable. When [*RT*] and *K_d_* are fixed, [*RL*] varies with [*LT*] and rises rapidly at first, and then tends to level off gradually, which is the saturation curve of receptor–ligand interactions. The equation indicates that the receptor–ligand interactions have a ligand saturation effect. This is similar to the interaction between enzyme and substrate, and the “substrate saturation effect” is a distinctive feature of enzymatic chemical reactions [25,26].

The hyperbolic curves obtained by fitting using the Origin9.0 software package are shown in Figure 5(A2–E2), with the equation for Figure 5(A2) as:(4)$$\Delta I=\frac{85.8981\times C\times 10^{-15}}{0.08908+C\times 10^{-15}}\left(R^{2}\;=\;0.9889\right)$$

At low concentrations, ΔI increased with increasing analyte (estrogens and estrogen mimics) concentrations, indicating that receptor binding sites were not saturated (Figure 5(A2–E2)). However, when the analyte concentration reached a threshold, ΔI remained unchanged or only slightly increased with further increases in analyte concentrations, indicating that the receptor binding sites were saturated. The above process, the hyperbolic fitting results, and the fitting formula suggested that the sensor reflects receptor–ligand interactions in addition to resultant signal amplification that mimics receptor–ligand interactions and signal transduction in vivo [25].

By modifying Equation (4), the following equation is obtained:(5)$$\frac{1}{\left[RL\right]}=\frac{1}{\left[RT\right]}+\frac{K_{d}}{\left[RT\right]}\frac{1}{\left[L\right]}$$

The above equation is a double reciprocal equation with 1/[*RL*] as the y-axis and 1/[*L*] as the x-axis. The slope of the line is *K_d_*/[*RT*], the intercept at the x-axis is −1/*K_d_*, and the intercept at the y-axis is 1/[*RT*]. Based on the above analysis, the reciprocal of estrogen concentration was plotted on the x-axis and the reciprocal of ΔI on the y-axis (i.e., a double-reciprocal plot) in Figure 5(A3–E3). The fitting linear regression equation is as follows:(6)$$\frac{1}{\Delta I}=0.00104\times 10^{-15}\times \frac{1}{C}+0.01168\left(R^{2}\;=\;0.9834\right)$$
(7)$$K_{d}=\frac{0.00104\times 10^{-15}}{0.01168}=8.904\times 10^{-17}\mathrm{mol}/L$$

According to the above equations, the Ka (*K_d_* in the aforementioned equation) value of E2 was 8.904 × 10^−17^ mol/L. Thus, the interconnected allosteric constants (Ka) for the other four ligand compounds were as follows: Ka, Res = 8.35 × 10^−16^ mol/L; Ka, G-1 = 8.00 × 10^−15^ mol/L; Ka, G-15 = 5.01 × 10^−15^ mol/L; Ka, BPA = 6.65 × 10^−16^ mol/L (Figure 6, columns with different colors indicate different GPER ligands). Smaller Ka values indicate more efficient amplification and signal output generation due to ligand–receptor interactions, thereby indicating higher detection sensitivity [27]. Among the five ligands, E2 exhibited the smallest Ka and the highest sensitivity. The sensor sensitivity of the sensors for the five ligands followed the order of E2 > BPA > Res > G-15 > G-1. Du et al. [28] developed a photoelectrochemical biosensor using an E2 aptamer as a recognition element to detect E2 with high sensitivity, achieving a detection limit of 3.3 × 10^−16^ mol/L under optimal conditions. The sensor fabricated in the present study achieved a detection limit one order of magnitude lower than that developed by Du et al. The reason that the GPER electrochemical receptor sensor prepared in this study exhibits such high sensitivity might be that the three-dimensional network structure formed after the chitosan (Chit) film can fix a large number of horseradish peroxidase (HRP) molecules and AuNPs. Specifically, the AuNPs with large specific surface areas and high conductivity adsorbed numerous GPER molecules through Au-S bonds. Importantly, the increased abundances of AuNPs, HRP, and active GPER molecules are key to the electrochemical signal amplification system and the extremely high sensitivity of the fabricated sensor [21].

The five GPER ligands exhibit similar structures (Figure 7A), including the presence of benzene rings in all ligands and phenol groups in BPA, Res, and E2. The sensor was most sensitive to E2 and the cyclopentanol moiety of E2 seems to play an important role in ligand–receptor recognition, as shown in the structural formula of Figure 7A. In addition, G-1 and G-15 also exhibit similar structures [12]. Compared with G-15, the acetyl group in G-1 is likely to induce conformational changes of GPER and trigger signal transduction through steric and/or polar environments of oxygen atoms.

### 2.6. Analysis of Molecular Simulation Docking Results of GPER with Estrogens and Estrogen Analogs

Affinity is an evaluation criterion for the degree of binding between the target receptor and small molecule ligands. The higher the absolute value of affinity, the stronger the ligand–receptor binding ability and the more stable the complex formed [29]. According to the affinity values of the optimal binding conformation of molecular docking, it is known that the affinity values of the binding conformation of GPER docked with resveratrol, estradiol and the other five ligand compounds are negative, which indicates that the reactions are all spontaneous. In other words, these estrogens and estrogen analogs can spontaneously bind to the active pocket of GPER without absorbing external energy. Subsequently, we observed the interactions within the complex at the amino acid level. The crystal structure of GPER and its binding sites to the five estrogens are shown in Figure 7G. The five estrogenic ligands interact with GPER at different binding sites, and Figure 7H shows the overlap of all ligands on GPER. Among them, resveratrol, bisphenol A, G-1 and G-15 overlap at one of the GPER binding sites, while E2 is located at the other GPER binding site.

When GPER docked with resveratrol at an affinity of −6.7 kcal/mol, the two showed hydrogen bonding interactions and the RMSD of resveratrol was 1.022. The residues Arg299 and Glu115 showed hydrogen bonding interactions with resveratrol with an interaction distance of 2.4 Å and 2.3 Å (Figure 7B). No pi–pi and pi–cation interactions were observed. The residues Gln54, Tyr55, Asn118, Leu119, Pro303, Gly306, His307, and Asn310 have hydrophobic interactions with resveratrol as shown by LigPlus analysis.

When GPER docked with bisphenol A with an affinity of −6.4 kcal/mol, the two showed hydrogen bonding interaction and the RMSD of bisphenol A was 4.655. The residue His300 had hydrogen bonding interaction with bisphenol A with an interaction distance of 2.6 Å (Figure 7C). There were no pi–pi and pi–cation interactions. The residues Gln54, Tyr55, Gly58, Leu119, His120, Pro303, and Leu304 have hydrophobic interactions with bisphenol A as shown by LigPlus analysis.

When GPER docked with 17β-estradiol at an affinity of −6.1 kcal/mol, the two showed hydrogen bonding interaction and the RMSD of estradiol was 3.816. The residue His282 showed hydrogen bonding interaction with estradiol with an interaction distance of 2.3 Å (Figure 7D). There were no pi–pi and pi–cation interactions. The residues Gln285, Arg286, Lys295, Gln296, Ser297, Phe298, and Ala301 have hydrophobic interactions with estradiol as indicated by LigPlus analysis.

When GPER docked with G-1 at an affinity of −6.5 kcal/mol, the two showed hydrogen bonding interactions and the RMSD of G-1 was 0.000. The residue Glu121 had hydrogen bonding interactions with G-1 with interaction distances of 2.2 Å (Figure 7E). The residues Tyr55, Leu119, His120, Cyx205, Cyx294 and His300 (pi–pi) have hydrophobic interactions with G-1 as shown by LigPlus analysis.

When GPER docked with G-15 at an affinity of −7.1 kcal/mol, the two showed hydrogen bonding interactions and the RMSD of G-15 was 0.000. The residue Asn310 had hydrogen bonding interactions with G-15 at an interaction distance of 2.4 Å (Figure 7F). There were no pi–pi and pi–cation interactions. The residues Gln54, Glu115, Asn118, Leu119, Glu 121, Tyr124, Arg299, Pro303, Leu304, and His307 have hydrophobic interactions with G-15 as shown by LigPlus analysis.

### 2.7. Molecular Dynamics Simulations

The dynamic behavior of the highest scoring compounds in receptor and molecular docking studies was analyzed using molecular dynamics simulations to investigate the stability of the binding conformation of the five estrogenic ligands upon binding to the GPER. We performed simulations of up to 100 ns for each system to explore the changes and conformational stability of the complexes of GPER proteins with the five estrogenic ligands. A number of properties were investigated, including hydrogen bond number, solvent accessible surface area (SASA), root mean square deviation (RMSD), root mean square fluctuation (RMSF), and radius of rotation (Rg).

The RMSD of the backbone atoms relative to the protein was calculated to assess the structural stability of the complex and to reflect the stability of the system during the simulation [30]. The root mean square deviation (RMSD) is obtained by using the “gmx rms” module. The RMSD plots (individual trajectories and combined trajectory files) (Figure 8A–E) show that the fluctuation pattern of 17β-estradiol (red) increased from 0 ns to 5 ns from 0 nm to 1.60 nm and fluctuated steadily from 5 ns to 16 ns. The RMSD increased gradually from 16 ns to 17 ns (1.60 nm to 2.30 nm) and decreased slightly from 17 ns to 18 ns. The RMSD fluctuated steadily from 18 ns to 40 ns and decreased slightly from 40 ns to 41 ns. From 41 ns to 42 ns, the RMSD gradually increased and fluctuated steadily from 42 ns to 90 ns. From 90 ns to the end of the trajectory, the RMSD decreased slightly. Almost all systems stabilized after 75 ns, except for the 17β-estradiol system, where fluctuations occurred at 90 ns. The RMSD of 17β-estradiol was stable at 1.75 Å; resveratrol, bisphenol A, and G15 stabilized at 0.75 Å, and G1 stabilized at 1 Å. The RMSD of 17β-estradiol is larger, indicating that it is less stable in binding to GPER. This is in line with the percussion theory that occurs when estrogen ligand compounds activate receptor conformational changes and produce physiological functions. The ligand does not bind completely to the receptor and the ligand compounds repeatedly knock the receptor, thereby increasing the efficiency of ligand activation of the receptor protein.

Conformational flexibility was analyzed by calculating the RMSF (Figure 9A–E). The RMSF plot was used to describe fluctuations in residue levels [30]. The RMSF results showed that the amino acid residues Gln53, Arg164, Arg251, Pro289, and Gly290 fluctuated more. Moreover, the fluctuations of Glu115 and Arg299 before and after GPER binding to Res, His300 in GPER interaction with BPA, His282 in GPER interaction with E2, Glu121 before and after GPER binding to G-1, and Asn310 before and after GPER binding to G-15 were small in the kinetic simulations at 100 ns. In the RMSF plots, the trends of the complexes indicate that the binding of these estrogenic ligands to the GPER is stable. The average RMSF values of the GPER-resveratrol, GPER-bisphenol A, GPER-17β-estradiol, GPER-G1, and GPER-G15 complexes are 0.188, 0.171, 0.162, 0.182, and 0.169 nm, respectively. The trend of the complexes in the RMSF plot indicates that the binding of these five estrogens to the GPER was stable. However, the lowest RMSF value of the GPER-17β-estradiol complex signifies higher stability among all the four complexes.

The solvent-accessible surface area (SASA) was used to determine the size of the protein volume expansion in each system (Figure 10A–E). Higher SASA values indicate an increase in protein volume with less fluctuation over the simulation time. The binding of ligands can alter SASA and sometimes have a significant effect on protein structure [30]. The averages SASA values of the GPER-resveratrol, GPER-bisphenol A, GPER-17β-estradiol GPER-G1, and GPER-G15 complexes were found to be 158.42 nm^2^, 158.98 nm^2^, 155.69 nm^2^, 158.65 nm^2^, and 157.59 nm^2^, respectively. The overall SASA analysis showed that the GPER-bisphenol A complex had the highest SASA value and the GPER-17β-estradiol complex had the lowest SASA value. From the overall observation, the binding of estradiol may reduce the expansion of the protein.

The compactness of the structure is expressed in terms of the radius of gyration (Rg). When fluctuations are small, the system is more compact and rigid, and remains consistent throughout the simulation [30]. We calculated the radius of gyration (Rg) values for GPER and the five estrogenic ligands to obtain the compactness of all the complexes (Figure 11A–E). The averages radius of gyration (Rg) of the GPER-resveratrol, GPER-bisphenol A, GPER-17β-estradiol, GPER-G1, and GPER-G15 complexes were found to be 2.110 nm, 2.084 nm, 2.064 nm, 2.094 nm, and 2.097 nm, respectively. The GPER-resveratrol complex had the highest Rg value, while the GPER-17β-estradiol complex obtained the lowest Rg value among all systems.

Hydrogen bonding analysis is an important parameter for the stability of protein ligands in molecular dynamics simulations [30]. In biochemistry, hydrogen bonding is an important interaction due to its vital role in molecular recognition, structural stability, enzyme catalysis, and drug distribution and permeability [30]. The solubility of a drug and its ability to establish important connections with biomolecular targets promotes strong binding and selectivity, which may be enhanced if it contains functional groups capable of generating hydrogen bonds. We calculated the number of hydrogen bonds formed during the 100 ns MD simulation to get an idea of the binding strength of the GPER and the five estrogenic ligands mentioned above (Figure 12). Compared to the other four estrogen ligand compounds, Res and G-1 produce more hydrogen bonds when bound to GPER, and E2 produces fewer hydrogen bonds when bound to GPER, making the GPER-E2 complex less stable, which is also in line with the percussion theory described above. Under the stimulation of molecular dynamics at 100 ns, E2 forms 1–2 hydrogen bonds with GPER.

Ka is defined as the substrate concentration at which half of the maximum response signal is reached when the ligand binds to the receptor. The change in conformation causes a change in the electrochemical parameters, which is ultimately measured as the rate of change in current. Different ligand compounds of the same concentration have different effects on conformation and produce different changes in electrochemical parameters. We believe that, the smaller the Ka value, the greater the change in conformation of the receptor protein activated by the same concentration of ligand compound and the greater the amount of change in electrochemical parameters. Corresponding to molecular dynamics simulations, the greater the distance change caused by the same concentration of ligand compounds.

Based on the above analysis, the conformational changes of GPER upon binding to the five estrogenic ligands were further investigated. To elucidate the possible relationship between Ka and the efficiency of ligands to promote GPCR into the active conformation, as well as the possible relationship with GPCR activity, we systematically analyzed the intracellular TM3-TM6 distance changes during MD simulation, including E2-Res (Figure 13B), E2-BPA (Figure 13C), E2-G1 (Figure 13D), and E2-G15 (Figure 13E). From the overall distance change analysis (Figure 13A), all five estrogenic ligand compounds can cause conformational changes in GPER, and Res, BPA, G-1, and G-15 do have estrogen-like functions and can perform the corresponding physiological functions in the actual in vivo environment. In addition, the TM3-TM6 distance after the binding of resveratrol, G1, and G15 to GPER (Figure 13B,D,E) was slightly reduced compared to E2, the natural substrate of GPER, and there was no significant difference in the TM3-TM6 distance after the binding of BPA to GPER (Figure 13C). The TM3-TM6 distance after the binding of Res, E2 and G-1 to GPER was stable, while the BPA distance decreased rapidly at 27–32 ns and increased after 32 ns; the TM3-TM6 distance of G-15 increased rapidly at 27 ns and decreased rapidly at 37 ns and was lower than the distance when GPER was bound to E2.

### 2.8. Stability and Reproducibility of Receptor Sensors

The sensor was used to measure the same concentration of E2 solution 10 times, resulting in an RSD of the current change rate of 5.81%. Thus, the sensor exhibited high stability. The receptor sensor prepared in different batches was tested using the same concentration of E2 solution. The RSD of the current change rate was 1.82%, suggesting that the receptor sensor had good stability. The receptor sensor was stored in ultrapure water at 4 °C and used to measure the same concentration of estradiol solution every day. The response current value of the sensor was relatively stable for the first nine days, and the response current value on day ten was 79.65% of that on day one, indicating that the sensor could be stored stably for at least nine days.

### 2.9. Discussion

In this study, an electrochemical GPER sensor was fabricated via self-assembly and was used to detect five GPER ligands. The sensor responded to the five GPER ligands and demonstrated a lack of strict selectivity. The GPER ligands include phytoestrogens that are active ingredients in almost all foods and Chinese herbal medicines in addition to exogenous estrogens used in industrial, agricultural, and pharmaceutical applications, as well as endogenous estrogens that can both interact with GPER, resulting in biological effects [4,31]. Among them, 17β-estradiol involved in this study is a natural class of steroids and is an essential estrogen. However, excessive amounts of 17β-estradiol and its metabolites may disrupt the endocrine system, lead to abnormal sexual function, and even induce cancer [32].

The sensing sensitivity of the sensor to the five ligands followed the order of E2 > BPA > Res > G-15 > G-1. Further, the sensor exhibited higher sensitivity to natural estrogens than exogenous estrogens. In addition, we considered the binding of the ligand compound to the outer (1–58) or inner (327–375) cellular region of the GPER, but the molecular docking results showed neither hydrogen bonding nor hydrophobic interactions with the outer cellular region, but rather hydrogen bonding in the inner cellular region (327–375) and membrane-embedded domain (59–87, 93–121, 129–160, 165–195, 212–244, 252–286, 300–326). Molecular simulation docking studies revealed that the residues Arg299 and Glu115 of GPER formed hydrogen bonds with the hydroxyl group (-OH) of Res, the residue His300 formed hydrogen bonds with the hydroxyl group (-OH) of BPA, the residue His282 formed hydrogen bonds with the hydroxyl group (-OH) of E2, the residue Glu121 formed hydrogen bonds with the imino group (-NH-) of G-1, and the residue Asn310 formed hydrogen bonds with the ether bond (-O-) of G-15. Compared with conventional estrogen detection methods [33], the sensor prepared in the present study demonstrated the highest sensitivity for five GPER ligands thus far observed (10^−17^ mol/L) and was significantly more sensitive than other methods, with detection time also being reduced to 60 s. These results collectively indicate that the sensor features advantages of high sensitivity, simple operation, fast detection, and high stability. Moreover, sensor measurements are independent of tissues or cells and mimic signaling cascades of cells as a novel signal transduction system is used. Thus, the electrochemical GPER sensor developed here can be used to detect estrogens and estrogen mimics.

The transmembrane transduction of signals from estrogen or from estrogen mimics through GPER is more analogous to enzymatic activity regulation rather than enzyme catalysis. Ligand recognition, allosteric modulation of GPER, and signal transduction occur over a short period of time; while enzymatic binding to substrates is required to convert substrates to products, estrogen–GPER interactions only require instantaneous allosteric modulation and activation of intracellular signaling cascades. Thus, this process can be better described as “tapping the keyboard” rather than “binding.” In other words, environmental signals transmit signals into the cells by “tapping” the receptor “keyboard” on cell membranes. The higher the signal transduction efficiency (sensitivity), the shorter the time of the receptor–ligand interactions and the higher the frequency of ligand “tapping” a receptor, as in the case of estrogen–GPER interactions. However, the kinetics of ligands interacting with GPERs in the G protein-coupled receptor superfamily have not yet been elucidated, nor have the fine structures of GPERs, necessitating further investigations of structure–function relationships.

## 3. Materials and Methods

### 3.1. Experimental Materials and Reagents

The expression vector PcDNA3.1(+) and hGPER gene (NM_001039966.1) were from Bioengineering (Shanghai) Co., Ltd., China; human-derived embryonic kidney cells (HEK-293T) from Shanghai Cell Bank, Chinese Academy of Sciences, China; Dulbecco’s modified Eagle medium (DMEM) and Penicillin mix—Penicillin mix from Gbico, (Grand Island, NY, USA); Lipofec-tamine 2000 was obtained from Invitrogen, Waltham, MA, USA; the whole protein extraction kit was purchased from Bioengineering (Shanghai) Co., Ltd., Shanghai, China; G-1, G-15, BPA, Resveratrol, and 17β-estradiol were from AbMole, Houston, TX, USA; Chloroauric acid (AuCl_4_H) was from Shenyang Jinke Reagent Factory, Shenyang, China; Trisodium citrate dihydrate (C_6_H_5_Na_3_O_7_) was from Tianjin Winda Rare Chemical Reagent Factory, Tianjin, China; Thionin acetate (Thi) was from Sigma, Santa Fe, NM, USA; Chitosan (Chit) (Deacetylation degree ≥ 90%) was from Jinan Haidebei Bioengineering Co., Ltd., Jinan, China; Horseradish peroxidase (~1000 units/mg) was from Sigma, Santa Fe, NM, USA; and the Ni-NTA Beadose Resin Kit was from Beijing Biolab, Beijing, China. All reagents are of analytical grade. Ultrapure water was used in this study.

### 3.2. Instruments and Equipment

Milli-Q Reference Ultrapure Water System (Shanghai Yarong Biochemical Equipment & Instruments Co., Ltd., Shanghai, China); CHI1040C electrochemical workstation and three-electrode system (glassy carbon electrode φ = 3 mm, reference electrode—Ag/AgCl electrode, counter electrode-platinum wire electrode) (Shanghai Brilliance Instruments Co., Ltd., Shanghai, China); UV-1800 UV-Visible Spectrophotometer (Japan Shimadzu Instruments Co., Ltd., Kyoto, Japan); the Transmission electron microscopy (TEM) (Philips Tecnai G2F20); and the Quanta FEG 250 scanning electron microscopy (SEM) (FEI, Hillsboro, OR, USA) were used in this study.

### 3.3. Statistical Analysis

The sensor data were analyzed and processed using OriginPro 2018 (Origin Lab, Northampton, MA, USA) and GraphPad Prism 9 (GraphPad Software, San Diego, CA, USA). All results are expressed as the mean ± standard deviation of at least three independent experiments. The experimental detection limit (LOD) was defined as the target concentration that gave a response current signal at least three times different from the standard deviation of the blank control signal.

### 3.4. Expression and Extraction of hGPER

The His-tagged hGPER-PcDNA3.1(+) glycerobacteria were expanded in LB medium, and then GPER recombinant plasmids were extracted using the Endotoxin Removal Plasmid Extraction Kit (operated according to the instructions). Next, the previously extracted plasmid was transfected into HEK-293T cells using the cationic liposome Lipofectamine 2000 transfection method (operated according to the instructions) and then continued to culture, and after 48 h of culture, GPER protein was extracted using the Whole Protein Extraction Kit (operated according to the instructions). We purified the proteins by using the Ni-NTA Beadose Resin Kit from Beijing Biolab.

### 3.5. Western Blot Analysis

The Western blotting workflow was used to complete the Western blotting analysis experiment. The samples were mixed with 5 × SDS-PAGE loading buffer in a volume of 1:4, placed in a 95 °C water bath for 5 min, followed by centrifugation at 12,000 r/min for 1 min, and the supernatant obtained was the electrophoresis sample. According to the manual of the molecular cloning experiment and the molecular weight of GPER protein, this electrophoresis was configured with 12.5% of separation gel and 5% of concentration gel. Next, 20 μL of sample and 5 μL of Rain Marker were put into the sample wells of the concentrated gel and electrophoresis was performed at a constant voltage of 80 V. After electrophoresis enters the separation gel interface, the voltage was adjusted to 120 V for electrophoresis. After the electrophoretic band was stopped at the end of the gel, the separation gel was cut off and transferred to PVDF membrane for semi-dry transfer at a constant voltage of 23 V for 1 h. After the transfer, the PVDF membrane was placed in TBST buffer (0.1 L TBS wash solution, add 1 mL Tween 20 then deionized water to concentrate to 1 L, and finally adjust pH to 7) and washed on a shaker for 5 min, then the PVDF membrane was removed and placed in 5% skim milk powder solution and closed at room temperature for 2 h. After closure, the PVDF membranes were washed with TBST buffer and then incubated overnight at 4 °C in 1:500 dilution of Anti-6×His antibody primary antibody. After the incubation was completed, the PVDF membranes were washed three times with TBST buffer for 20 min each, and then washed with TBS buffer for 10 min. After washing, the membranes were removed and placed in HRP-goat anti-rabbit secondary antibody diluted 1:1000 and incubated at room temperature for 2 h. After completion of incubation, the membranes were washed three times with TBST buffer for 15 min each time. Then the membrane was washed with TBS buffer for 10 min. Finally, the PVDF membranes were removed and put into the developing solution, and the results were analyzed by developing under a fully automated chemiluminescence imaging analysis instrument.

### 3.6. Preparation and Detection of the hGPER Electrochemical Receptor Sensor

#### 3.6.1. Pre-Treatment of Glassy Carbon Electrodes

The glassy carbon electrode (GCE) was polished using aluminum powder (α-Al_2_O_3_) on chamois and then rinsed with ultrapure water, then cleaned with ultrasonic water bath oscillation for 30 s and repeated 3 times, and then the treated electrode surface was cleaned with 50% HNO_3_, anhydrous ethanol, and ultrapure water in turn.

After cleaning, the glassy carbon electrode was activated by cyclic voltammetry (scan rate of 100 mV/s) through H_2_SO_4_ using a three-electrode system (Ag/AgCl electrode as the reference electrode and platinum wire electrode as the counter electrode) with reference to the method in [23]. After treating the electrode, the electrode was placed in 1 × 10^−3^ mol/L K_3_Fe(CN)_6_ solution containing 0.20 mol/L KNO_3_ and scanned by cyclic voltammetry with a scanning range of −0.2 to 0.4 V and a scanning speed of 50 mV/s. When the redox peak potential difference is less than 80 mV, the GCE pretreatment is successful; otherwise, it should be re-polished. After scanning, the electrode was rinsed with ultrapure water and dried in a nitrogen atmosphere before it could be used for experiments.

#### 3.6.2. Preparation and Characterization of AuNPs

First, the glassware such as triangle bottles and volumetric bottles needed in the preparation process were cleaned and dried for use. Trisodium citrate was used to reduce chloroauric acid for preparing AuNPs, following the Frens method [34]. The specific method is as follows: mix 0.01 g/100 mL of chloroauric acid solution with 1 g/100 mL of sodium citrate solution in a ratio of 25:1 (Chloroauric acid solution 100 mL, sodium citrate solution 4 mL), and adjust the pH of the solution to 7.0 with K_2_CO_3_ and Na_2_CO_3_. The mixture was heated in the microwave oven at high temperature, and the color change of the solution was observed at any time until the bright red wine color appeared, and the AuNP sols were successfully prepared. The prepared AuNP sols were scanned in the visible range (400–700 nm) using a UV-Vis spectrophotometer, and the size of the prepared AuNP sol particles was roughly characterized by the wavelength at the maximum absorption peak. Subsequently, the size, shape, and dispersion of the prepared AuNP sols particles were further characterized precisely by transmission electron microscopy.

#### 3.6.3. Preparation of Double-Layer hGPER Receptor Nanogold Sensor

Referring to the literature and with slight improvements [25], the pretreated and characterized GCE surface was coated dropwise with 5 μL of 0.5% chitosan (Chit) solution (Chit was dissolved in 1% acetic acid solution), dried at 37 °C, and the drying was stopped after the appearance of Chit film (about 30 min) to characterize the electrode. After the electrodes were cooled to room temperature, the electrodes were immersed in 0.5 mol/L NaOH solution for 5 min, then washed with ultrapure water and characterized, and then the electrodes were immersed in nanogold sols (GNPs) and self-assembled at 4 °C over-night, removed, and washed with ultrapure water and characterized. Afterwards, 5 μL of GPER receptor protein was added dropwise to the electrode, self-assembled, and adsorbed at 4 °C for 12 h. The electrode was washed with ultrapure water and then characterized. We applied 5 μL of the Thi-Chi complex (160 μL of 10% glutaraldehyde was added to 1.25 mL of 2% chitosan solution (chitosan was dissolved in 1% acetic acid solution), mixed and added 100 μL of 0.01 mol/L Thi-Chi solution, and finally fixed to 3 mL with 2% acetic acid solution, mixed and used 5 μL) dropwise on the electrode surface, and then naturally after drying, the electrode was characterized. Then the electrode was immersed in nanogold-horseradish peroxidase (HRP-GNPs) mixture solution at 4 °C for self-assembly overnight. (The HRP-GNPs mixture was obtained by mixing 1 mL of GNPs obtained above with 1 mL of 2.0 g/L HRP (essentially salt-free, lyophilized power 250 units/mg solid, and dissolved in 0.01 M PH = 7.4 PBS buffer) for two hours and left to stand for 12 h at 4 °C.) We then removed the electrode, washed it with ultrapure water, and characterized it. The electrode was coated with 5 μL GPER receptor protein in a second drop, self-assembled, and adsorbed at 4 °C for 12 h. The electrode was washed with ultrapure water and then characterized. Finally, 5 ul of 0.5% bovine serum protein (BSA) solution was applied dropwise and incubated at 37 °C for 1 h to seal the non-specific sites, and the electrode was removed and washed with TBS solution and then rinsed with ultrapure water to characterize the electrode. At this point, the preparation of the bilayer nanogold modified GPER electrochemical receptor sensor was completed. The ligands (i.e., G-15, G-1, estradiol, bisphenol A, and resveratrol in this experiment) interact with the GPER on the electrode to generate a weak electrical signal (caused by a change in spatial site resistance that prevents electron transfer), which can be collected by an electrochemical workstation. See Figure 8 for the GPER electrochemical receptor sensor assembly process.

### 3.7. Measurement with the Electrochemical GPER Sensor

A three-electrode system was used, with a glassy carbon electrode with a measuring membrane as the working electrode, a Ag/AgCl electrode as the reference electrode, and a platinum wire electrode as the counter electrode, and ultrapure water was used as the blank control. The response currents of the GPER electrochemical receptor sensors to different concentrations of GPER ligands (estradiol, bisphenol A, resveratrol, G-15, G-1) were measured in parallel three times at each concentration. The equation for calculating the rate of change of response currents was as follows:(8)$$\Delta I\;/\;\%\;=\frac{I_{1}-I_{2}}{I_{1}}\times 100\Delta I\;/\;\%\;=\frac{I1-I2}{I1}\times 100$$
where ΔI is the rate of change for response current, I_1_ is the response current of the control electrode, and I_2_ is the response current of GPER ligands (E2, Res, BPA, G-1, and G-15).

The kinetics of signaling cascades triggered by GPER ligand–GPER interactions mimic the kinetics of enzymatic reactions [25]. In this study, the linear range of ligand concentrations was determined and subdivided based on current changes generated by the interaction between different concentrations of ligands and GPERs. Analysis of the change rate of the response current demonstrated that different GPER ligand–GPER interactions met the hyperbolic fitting criteria (R^2^ ≥ 0.95). Furthermore, the interconnected allosteric constant (Ka) of ligands was calculated. A diagram showing the principles of measurement for the entire assembly of GPER sensor is shown in Figure 14. GPER proteins were assembled onto the electrode as a biorecognition element to measure the solutions of five GPER ligands. An electrochemical signal amplification system was then used to simulate intracellular receptor signaling cascades to achieve ultrasensitive measurements of signal amplification that were triggered by interactions of estrogens or estrogen mimics with GPER, followed by further analysis of reaction kinetics.

### 3.8. Molecular Simulation Docking of GPER with Estrogens and Estrogen Analogs

#### 3.8.1. Preparation and Handling of GPER with Estrogens and Estrogen Analogs

The crystal structures of the desired proteins Homo sapiens G protein-coupled estrogen receptor 1 (GPER1), transcript variant 3, mRNA can be searched in the AlphaFold protein structure database by amino acid sequence matching. The structures of the desired estrogens and estrogen analogs can also be searched in the PubChem database.

The receptor protein and ligand need to be processed before molecular docking. First, we used AutoDockTools to remove the water molecules, added all hydrogen atoms of GPER, and use them as molecular docking acceptors, and we added all hydrogen atoms of each of the estrogens and estrogen analogs and used them as molecular docking ligands. Next, the ligands were examined for torsional bonding.

#### 3.8.2. Molecular Docking Parameter Settings

We used AutoDockTools to determine the parameters of the active pocket. Next, the 3D structure of the ligand needed to be contained in the Gridbox in its entirety. Therefore, the molecular docking parameters of GPER with the five estrogens and estrogen analogs are shown in Table 1. All other parameters are default values.

#### 3.8.3. Molecular Simulation Docking of GPER with Estrogens and Estrogen Analogs

We investigated the molecular docking mechanism of GPER with estrogens and estrogen analogs using the AutoDock Vina 1.5.6 software. The AutoDock Vina program continuously adjusted the conformations of ligand molecules within the Gridbox and then scored the different conformations of ligands (including orientation, position, energy, etc.). During the docking process, each of the estrogens and estrogen analogs produced nine docked conformations, and each conformation was ranked from highest to lowest affinity value. The conformation with the highest affinity value indicated the best geometric and energetic match between the ligand molecule and the receptor protein. Subsequently, the binding patterns and interactions of GPER with the estrogens and estrogen analogs were analyzed by PyMol (The PyMOL Molecular Graphics System, Version 2.4 Schrodinger, LLC.) and LigPlus v.2.2.8.

### 3.9. Molecular Dynamics Simulations

The five GPER ligands were optimized at the B97-3c level using ORCA [35], and then single point energy was calculated at the B3LYP-D3(BJ)/def2-TZVP level to simultaneously generate molecular surface electrostatic potential data, and finally Multiwfn [36] was called to generate RESP charges [37], which were combined with the small molecule topology generated in the sobtop [38] file for binding. Through Gromacs2022.1, the amber14sb_parmbsc1 force field was used for the protein model and the TIP3P model was adopted for the water model, with a minimum distance of 1 Å between the solute atom and the edge of the periodic box, as well as energy minimization using the fastest descent method, set to a maximum of 1000.0 kJ·mol^−1^·nm^−1^. The equilibration of the system was carried out by 100 ps restricted kinetics with a step size of 1 fs. Constant temperature and pressure simulations of the system were carried out using Velocity-rescale and Berendesen, with the temperature set to 298.5 K and the pressure set to 1.01325 bar. The Particle-Mesh Ewald (PME) method was used to deal with long-range interactions, and the Van der Waals interaction was truncated at 10 Å. As the ligand–protein receptor interactions were more pronounced, they were confined together in the task as the same temperature-controlled group and the same group that eliminates translational rotation, followed by conventional molecular dynamics simulations of the three systems in steps of 2 fs and 100 ns duration, with a coordinate file recorded every 10 ps.

Five commonly used methods—Root Mean Square Deviation (RMSD), Root Mean Square Fluctuation (RMSF), radius of gyration (Rg), solvent accessible surface area (SASA), and number of hydrogen bonds—were used to analyze protein stability and parameter changes in kinetic processes. After the molecular dynamics simulation of the protein ligand trajectory were completed, periodic calibration and stability analysis of the simulated trajectory were performed to calculate the mean and standard deviation of the RMSD to assess the stability of the protein ligand system, in combination with Rg, SASA, and hydrogen bond number to assess whether the conformation of the inhibitor had changed the conformation of the enzyme.

## 4. Conclusions

This study is the first to use GPER to prepare an electrochemical double-layer AuNP-modified GPER sensor. The electrochemical workstation and the GPER electro-chemical receptor sensor were used for the quantitative detection of five important GPER ligands and calculation of kinetic parameters. The resulting sensitivity of the sensor for the five ligands followed the order of E2 > BPA > Res > G-15 > G-1. Further, the sensor was more sensitive to natural estrogens than exogenous estrogens. In addition, we further performed molecular simulation docking of GPER with five major GPER ligand compounds to analyze their binding modes and interactions. The results showed that the residues Arg, Glu, His, and Asn of GPER mainly formed hydrogen bonds with -OH, C-O-C, or -NH- of the estrogens and estrogen analogs. Compared with conventional estrogen detection methods, the electrochemical GPER sensor exhibited the advantages of high sensitivity, high reproducibility, low cost, simple operation, and fast detection. The sensor also overcame problems associated with short lifetimes of typical cell-based biosensors [39]. Overall, the electrochemical signal amplification system developed in this study was used to mimic receptor signaling cascades in cells, thereby helping to directly study the interactions between GPER and ligands while exploring sensing kinetics. These results provide the theoretical basis for the future accurate functional evaluation of food ingredients or toxins from the receptor perspective. However, the purified protein GPER was used in this study and its binding to the five estrogenic ligands mentioned above is not identical to the binding of GPER to estrogenic ligands in vivo, and this study is meant only to provide an idea for research.

## Figures and Tables

**Figure 1 molecules-28-03286-f001:**
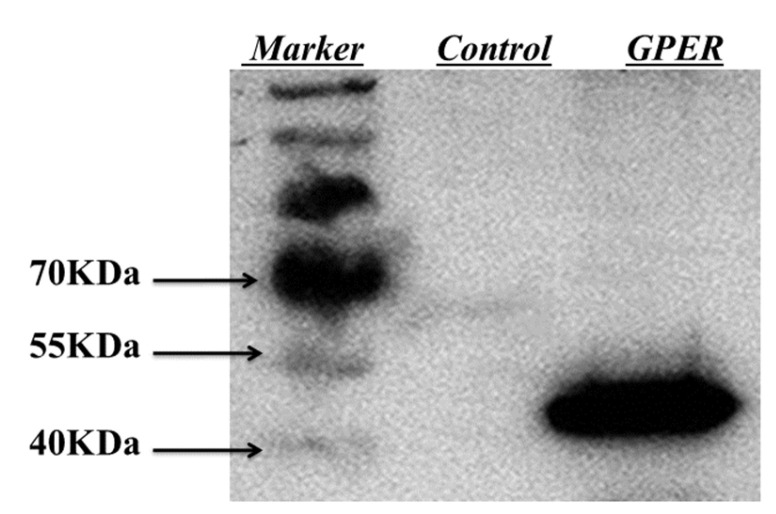
Western blotting of GPER; Characterization of the effect of the GCE pretreatment.

**Figure 2 molecules-28-03286-f002:**
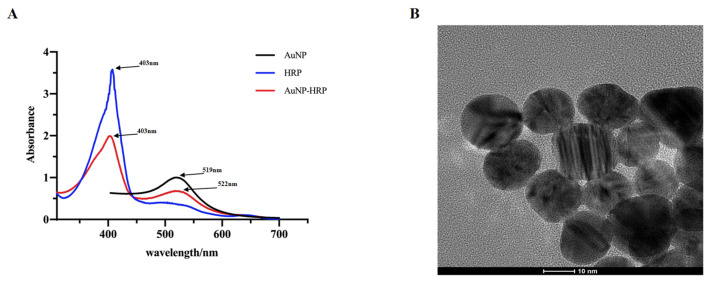
(**A**) UV-Vis Spectroscopic scanning characterization of GNPs and the effect of GNPs adsorption of HRP; (**B**) TEM representation images of the configured GNPs.

**Figure 3 molecules-28-03286-f003:**
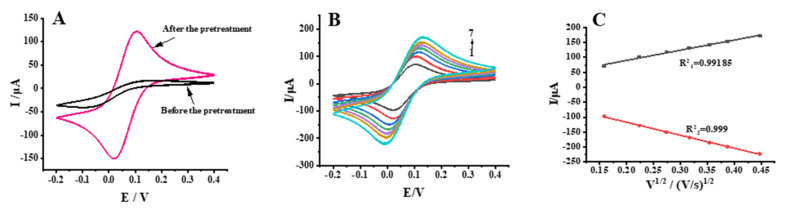
(**A**) Cyclic Voltammetry before and after bare electrode pretreatment; (**B**) Cyclic voltammograms of bare GCE at scan rates of 0.025, 0.050, 0.075, 0.100, 0.125, 0.150, 0.200 V/s (1−7); (**C**) The inset shows the dependence of the redox peak currents on the square root of scan rates.

**Figure 4 molecules-28-03286-f004:**
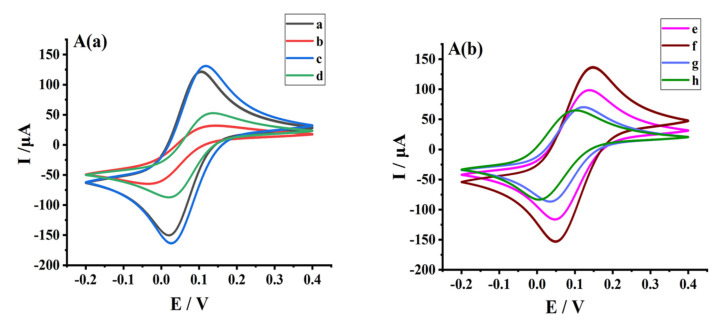
The cyclic voltammetry characterization of the sensor assembly process. The sensor assembly process is as follows: (**A**(**a**)) (a−d) GCE, Chit−GCE, GNPs−Chi−GCE, GPER−GNPs−Chit−GCE, (**A**(**b**)) (e−h)Thi/Chit−GPER−GNPs−Chit−GCE, HRP/GNPs−Thi/Chit−GPER−GNPs−Chit−GCE, GPER−HRP/GNPs−Thi/Chit−GPER−GNPs−Chi−GCE, and BSA−GPER−HRP/GNPs−Thi/Chit−GPER−GNPs−Chit−GCE. Cyclic voltammetry was carried out in a K_3_Fe(CN)_6_ solution (1 × 10^−3^ mol/L, containing 0.20 mol/L KNO_3_) with a scan rate of 50 mV/s and a scan range of −0.2 to 0.4 V, using a three−electrode system: a glassy carbon electrode with a measuring membrane as the working electrode, an Ag/AgCl electrode as the reference electrode, and a platinum wire electrode as the counter electrode.

**Figure 5 molecules-28-03286-f005:**
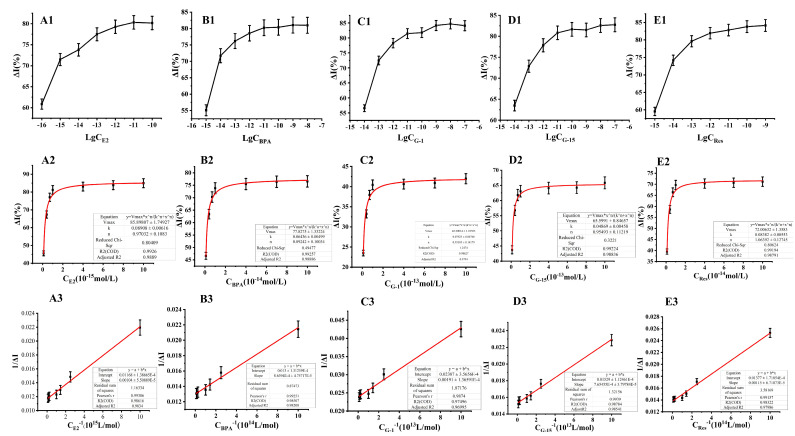
(**A1**) Current response values of E2 in the detection range (C = 10^−16^ mol/L~10^−10^ mol/L); (**A2**) The curve of interaction between E2 (10^−16^ mol/L~10^−14^ mol/L) and the receptor; (**A3**) Plotting with double reciprocal: E2 in the concentration range of 10^−16^ mol/L~10^−14^ mol/L; (**B1**) Current response values of BPA in the detection range (C = 10^−15^ mol/L~10^−8^ mol/L); (**B2**) The curve of interaction between BPA and the receptor; (**B3**) Plotting with double reciprocal of BPA; (**C1**) Current response values of G−1 in the detection range (C = 10^−14^ mol/L~10^−7^ mol/L); (**C2**) The curve of interaction between G−1 and the receptor; (**C3**) Plotting with double reciprocal of G−1; (**D1**) Current response values of G−15 in the detection range (C = 10^−14^ mol/L~10^−7^ mol/L); (**D2**) The curve of interaction between G−15 and the receptor; (**D3**) Plotting with double reciprocal of G−15; (**E1**) Current response values of Res in the detection range (C = 10^−15^ mol/L~10^−9^ mol/L); (**E2**) The curve of interaction between Res and the receptor; (**E3**) Plotting with double reciprocal of Res.

**Figure 6 molecules-28-03286-f006:**
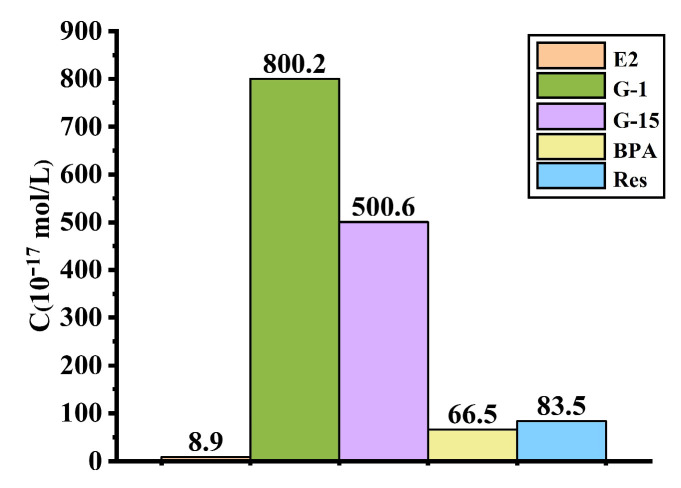
Interconnected allosteric constants of estrogens and their analogs determined by a self-assembled double-layer gold hGPER electrochemical receptor sensor.

**Figure 7 molecules-28-03286-f007:**
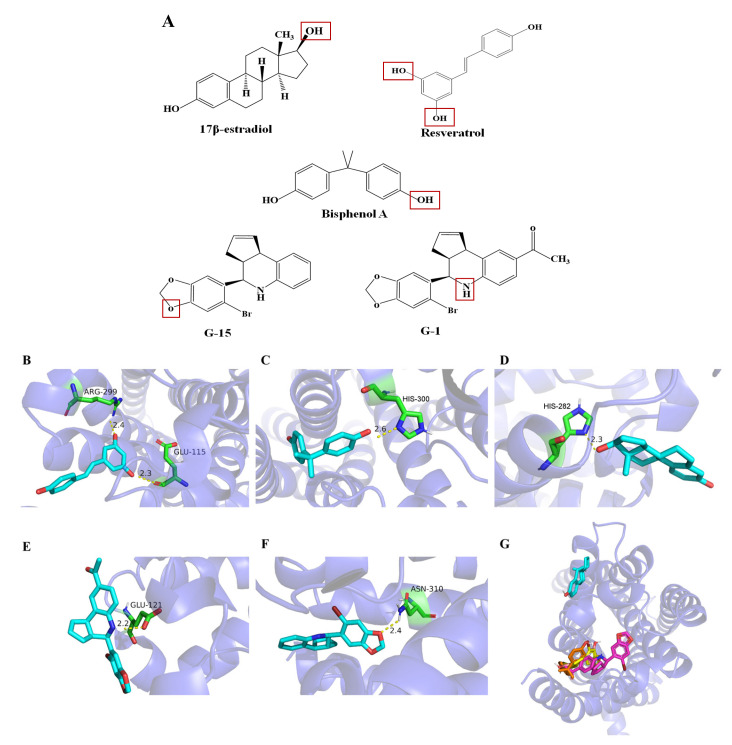
(**A**) Structures of GPER ligand compounds; 3D schematic diagram of molecular docking between GPER and five estrogens and their analogs: (**B**) The residues Arg299 and Glu115 showed hydrogen bonding interactions with resveratrol with an interaction distance of 2.4 Å and 2.3 Å; (**C**) The residue His300 had hydrogen bonding interaction with bisphenol A with an interaction distance of 2.6 Å; (**D**) The residue His282 showed hydrogen bonding interaction with 17β-estradiol with an interaction distance of 2.3 Å; (**E**) The residue Glu121 had hydrogen bonding interactions with G-1 with interaction distances of 2.2 Å; (**F**) The residue Asn310 had hydrogen bonding interactions with G-15 at an interaction distance of 2.4 Å; (**G**) Crystal structure of GPER and its binding sites with five estrogens and estrogen analogs: resveratrol (salmon); bisphenol A (orange); 17β-estradiol (cyan); G-1 (magenta); G-15 (yellow).

**Figure 8 molecules-28-03286-f008:**
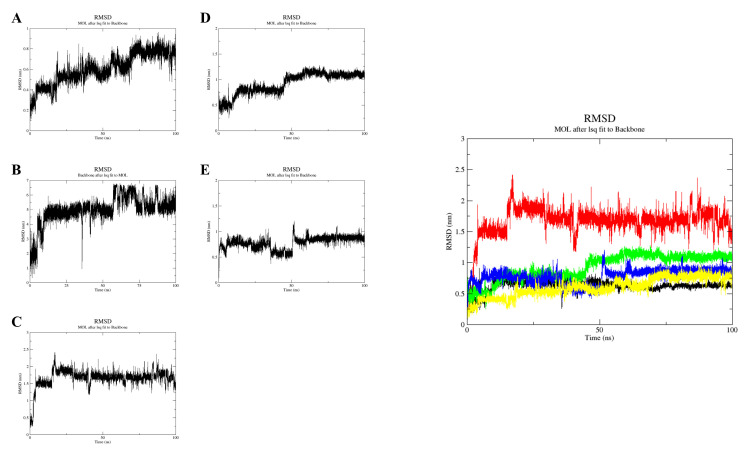
RMSD plot (individual trajectory) of resveratrol (**A**); bisphenol A (**B**); 17β-estradiol (**C**); G-1 (**D**); G-15 (**E**). RMSD plot (merged trajectories) of resveratrol (yellow); bisphenol A (black); 17β-estradiol (red); G-1 (green); G-15 (blue).

**Figure 9 molecules-28-03286-f009:**
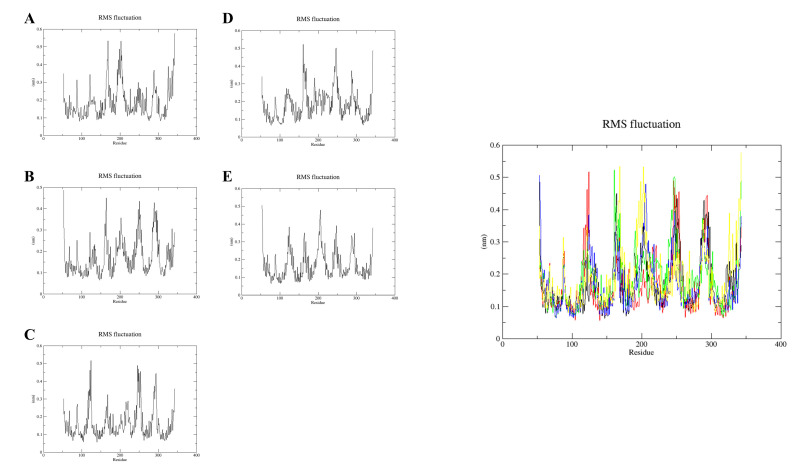
RMSF plot (individual trajectory) of resveratrol (**A**); bisphenol A (**B**); 17β-estradiol (**C**); G-1 (**D**); G-15 (**E**). RMSF plot (merged trajectories) of resveratrol (yellow); bisphenol A (black); estradiol (red); G-1 (green); G-15 (blue).

**Figure 10 molecules-28-03286-f010:**
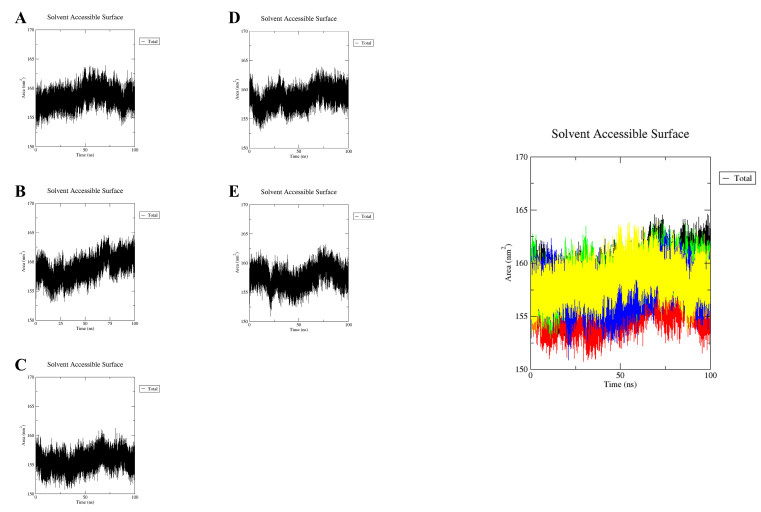
Estimation of SASA values of resveratrol (**A**); bisphenol A (**B**); 17β-estradiol (**C**); G-1 (**D**); G-15 (**E**). Estimation of SASA values of resveratrol (yellow); bisphenol A (black); estradiol (red); G-1 (green); G-15 (blue).

**Figure 11 molecules-28-03286-f011:**
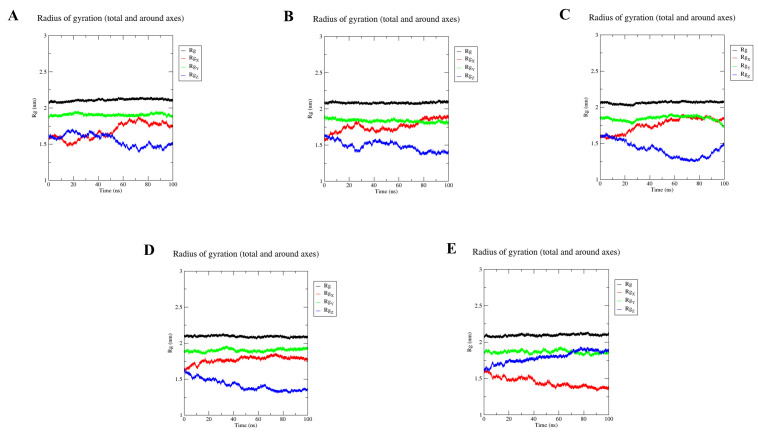
Radius of gyration (Rg) plot of resveratrol (**A**); bisphenol A (**B**); 17β-estradiol (**C**); G-1 (**D**); G-15 (**E**).

**Figure 12 molecules-28-03286-f012:**
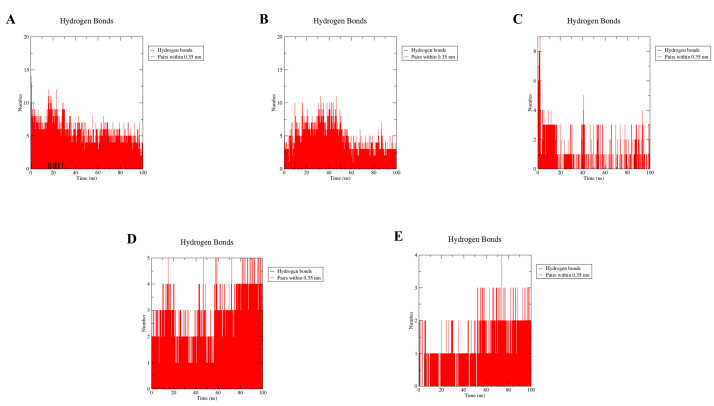
Analysis of hydrogen bond numbers of resveratrol (**A**); bisphenol A (**B**); 17β-estradiol (**C**); G-1 (**D**); G-15 (**E**).

**Figure 13 molecules-28-03286-f013:**
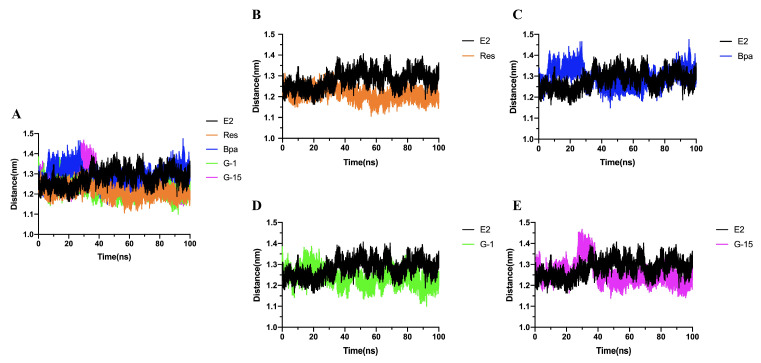
(**A**) The changes of intracellular TM3-TM6 distance during 100 ns MD simulations were calculated for resveratrol (orange); bisphenol A (blue); 17β-estradiol (black); G-1 (green); G-15 (magenta). Comparison of changes in TM3-TM6 distance upon binding of Res (**B**), BPA (**C**), G-1 (**D**), G-15, and (**E**) GPER with E2 and GPER.

**Figure 14 molecules-28-03286-f014:**
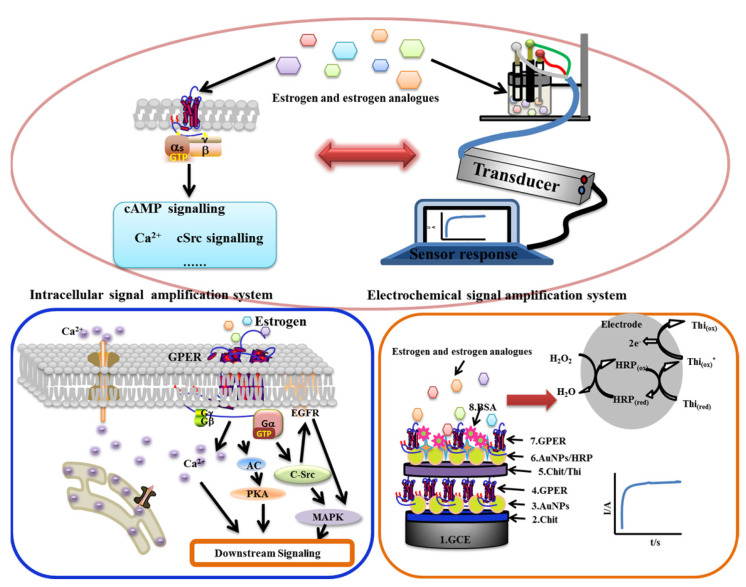
Schematic diagram of measuring principle of GPER electrochemical receptor sensors.

**Table 1 molecules-28-03286-t001:** Molecular docking parameters between GPER and five estrogens and estrogen analogs.

	Gridbox Center Coordinates			Gridbox SIZE		
	Center_x	Center_y	Center_z	Size_x	Size_y	Size_z
RES	5.542	2.038	−13.944	30.0	25.5	33.0
BPA	5.542	2.038	−13.944	30.0	25.5	33.0
E2	5.542	2.038	−13.944	30.0	25.5	33.0
G1	5.542	2.038	−13.944	30.0	25.5	33.0
G15	5.542	2.038	−13.944	30.0	25.5	33.0

## Data Availability

This article has not been submitted to other journals, and the cited materials are labeled references.

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
