# Peer review of "Study of the Sensing Kinetics of G Protein-Coupled Estrogen Receptor Sensors for Common Estrogens and Estrogen Analogs"

_molecules, 2023, doi:10.3390/molecules28083286_

Round 1
Reviewer 1 Report
This manuscript titled “Study of the sensing kinetics of G protein-coupled estrogen receptor sensors for common estrogens and estrogen analogues” by DingQiang Lu et al. applied a electrochemical sensor to measure the sensing kinetics for estrogen compounds, and the results show the difference in allosteric constants (Ka) for selected ligands, they also try to study the mechanism using docking. The electrochemical sensor part of this study is very interesting, however, I have a lot of concerns on the molecular docking part of this work. The senor part of this study reveals the kinetic differences in the selected ligands. It is very important to continue study the structural mechanism behinds the kinetic differences to further our understanding. The molecular docking part was meant to serve that purpose, but failed to do so, because the design of the docking part is not careful, and the data collected was not solid. My major concerns are listed below:
1. Alpha Fold predicted structure was used as grid. However, the accuracy of the Alpha Fold model have to be very carefully investigated. How much trust can we put into the Alpha Fold predicted structure? Does Alpha Fold has high confidence level around the ligand binding site? Even it has high confidence, how to prove it is accurate enough to be used as docking template? I highly appreciate the effort from Alpha Fold team but to use their structure without validation is not a solid approach.
2. Figure 8 is not clear and very confusing. I can not see the relative binding pose of ligand in the GPCR. Turning on shadow in Pymol and ray it will help. Some of the ligand even seems to bind in different binding pocket, for example, the ligand in panel B seems to bind from membrane side. The structure figures need more careful tune to show the message.
3. The main results from the docking part is mostly docking energy and a description of contact between ligand and GPCR. However, docking is a quick way to give an estimation of how ligand will bind, docking itself is not considered as solid evidence to provide mechanism insight, except supported by mutagenesis experiment. One possible solution is to perform molecular dynamics simulation from docking pose to study the dynamics interaction between ligand and GPCR. Important ligand contacting residues can be identified from MD simulation, then, mutagenesis on the critical residues shall be performed. Docking, MD, mutagenesis together should offer solid insight into the mechanism. Currently, this study only involve docking, which is problematic to build any conclusion on.
4. If mutagenesis is not doable, alternative approach should be taken to shed light in the mechanism. I am thinking of NMR but not sure if it is a proper tool for this GPCR.
5. I don’t suggest fully remove the mechanism section. Because this manuscript is very similar to ref 16, only switching GPCR and tested ligands. This manuscript will lack in novelty if the mechanism insight is missing.
Overall, I think this manuscript need a major revision to strengthen the mechanism part.
Author Response
Please see the attachment.
TRANSLATE with x English
| Arabic | Hebrew | Polish |
| Bulgarian | Hindi | Portuguese |
| Catalan | Hmong Daw | Romanian |
| Chinese Simplified | Hungarian | Russian |
| Chinese Traditional | Indonesian | Slovak |
| Czech | Italian | Slovenian |
| Danish | Japanese | Spanish |
| Dutch | Klingon | Swedish |
| English | Korean | Thai |
| Estonian | Latvian | Turkish |
| Finnish | Lithuanian | Ukrainian |
| French | Malay | Urdu |
| German | Maltese | Vietnamese |
| Greek | Norwegian | Welsh |
| Haitian Creole | Persian |
TRANSLATE with
EMBED THE SNIPPET BELOW IN YOUR SITE
Enable collaborative features and customize widget: Bing Webmaster Portal Back

Reviewer 2 Report
Comments and suggestions for authors
This is an interesting manuscript, with several strengths. This article fits the scope of the journal and provides more data on the importance to evaluate the presence of food-functional ingredients or toxins with estrogen-like effects which can lead to endocrine disorders and even cancer in humans. In order to straighten this article, points should be addressed accordingly before consideration in the journal:
Title
- I think there is a mistake in the title, “analogues” should be “analogs” please tell me if is it correct!
Abstract
-The abstract is a bit short. Please rewrite the abstract by providing the research design in the materials and methods section and the results section should be more detailed.
Introduction
-The author should provide the objective/aim of the study in the last paragraph of the introduction!
-Also indicate your experimental approach!
-Why is your experimental approach new, different, and important? Novelty!
Materials and methods
-Please indicate the brand and country of all chemicals/compounds used in this work.
-Please separate units from numbers throughout the manuscript.
I saw that there was no statistical analysis section here, did you perform a statistical analysis? Please provide it to enhance the significance of your results!
-I did not find any references in the first sections of the material and methods, please provide them!
Results and discussion
-In general, figures and tables should be understandable without reading the manuscript itself. Therefore, please consider explaining them briefly in the footnotes.
-Please indicate the significant statistic in all tables and figures.
-For example, in Figure 6 you have indicated the standard deviation, but you have not mentioned how the statistical analysis was performed.
-Is there a limitation or weakness of the study?
Conclusion
-Please consider rewriting the conclusion, it is too long. Restate your main finding and give a final take-home message.
Author Response

(The authors gave the same response as above.)

Reviewer 3 Report
The manuscript about Study of the sensing kinetics of G protein-coupled estrogen re- 2 ceptor sensors for common estrogens and estrogen analogues.
The organization of the manuscript is good and the presenting of data are perfect.
It could be accepted after minor comments:
1- English language should be revised by a native English speaker as there are some gramatical mistakes.
2- Figure 8 resolution are very low, author have to split it into two figures to increase resolution.
3- More data and details should be added to the abstract.
Author Response

(The authors gave the same response as above.)

Round 2
Reviewer 1 Report
I appreciate the authors’ efforts in preparing MD date to address my previous comments. Here I have more detailed comments which I hope can offer a more solid mechanism insight to the experiment.
1. The new figure 7 is clearer now, but there are few improvements that can be made. All the hydrogen atoms can be hidden. The atoms should be colored by element using PyMol color pull down menu, right now the His are all green and I can’t see which atom is nitrogen. HIE is CHAMM FF name, and not so friendly to general audience. Please change all HIE to HIS.
2. The purpose of MD simulations is to offer mechanism insight to experiment observation. The measurement done in the current manuscript, including hydrogen bond, SASA, RMSD, RMSF, Rg, are in general considered as quality check measurement for MD, which means they can only tell us whether the simulations are good enough for publication or not. Can these measurements offer us mechanism insight? It depends on the question waiting for answer, but most of the time they are not relevant.
3. The first section comes to two conclusions: first is the Ka values for the five ligands, second is the sensitivity of sensor for the five ligands. I think the MD should provide some insights that are relevant to these two conclusions. I have to admit I’m not familiar with the technic used in the first section, but if my understanding is correct, the Ka is related to the efficiency of ligand promote GPCR into active conformation. If this is true, then some measurements related to GPCR activity should be more relevant. For example, you can measure the intracellular TM3-TM6 distance and TM3-TM7 distance, which is considered as signature of activation. Or you can measure the distance between OH atoms in the Y-Y motif. Some structure papers from Kobilka or Christ Tate, or some computational paper from Ron Dror will lend some ideas on what to measure. The point here is to show that the ligands with higher (or lower?) Ka has better ability to open up the G protein binding site. If the distance differences are not clear in the current 100ns simulation, then longer simulations are needed.
4. If the previous distance measurement agrees with experiment, then the next question is why the five ligands trigger different responses. To answer that question, it’s necessary to look at the ligand-GPCR interaction, and maybe also ligand stability.
5. The current method section for the five analysis is too simple, I can’t replicate what has been done. For example, the RMSD was done on protein backbone? Or Ca atoms only? Or for the ligand heavy atoms? I don’t know the RMSD values in Figure 8 are calculated from which part of the system. I would think RMSD of ligand will be helpful since it explains which ligand form stable binding with GPCR.
6. The H-bond analysis may also help explaining the differences in the five ligands, but currently there are only five panel in Figure 12, without any interpretations. What is the main conclusion from figure 12? Which ligands binds best? How to use a single value to show a quantitative comparison among the five ligands? One common method besides counting H-bound will be the contact frequency from MD trajectory. This can be done easily using get_contact (https://getcontacts.github.io/).
7. The MD analysis need to combine with the docking results. There are several residues identified and discussed in the docking section. Are those contact remains during MD? What are the roles of those contact in MD? And how to use them to explain the differences observed in Ka?
Overall, the major problem from the current version is that the docking and MD section seems to be independent from the kinetic measurement. They didn’t offer any insight or related to the Ka values. The figures in the docking and MD section are only a stacking of raw data without implementing into the main story. Several figures don’t even contain any meaningful data, or the data was not fully discussed in the text.
Author Response

(The authors gave the same response as above.)

Reviewer 2 Report
Thank you for your response. In order to straighten this article, points should be addressed accordingly before consideration in the journal:
Materials and methods
I saw that there was no statistical analysis section here, did you perform a statistical analysis? Please provide it to enhance the significance of your results!
Rewrite the statistical analysis, it should be more detailed, please check other papers to have an idea of how to do it.
Author Response

(The authors gave the same response as above.)
